# TalkToModel: Explaining Machine Learning Models with Interactive Natural Language Conversations

**Dylan Slack**
University of California, Irvine
dslack@uci.edu

**Satyapriya Krishna**
Harvard University
skrishna@g.harvard.edu

**Himabindu Lakkaraju**[*]
Harvard Univesity
hlakkaraju@hbs.edu

**Sameer Singh**[*]
University of California, Irvine, Allen AI
sameer@uci.edu

## Abstract

Machine Learning (ML) models are increasingly used to make critical decisions in real-world applications, yet they have become more complex, making them harder to understand. To this end, researchers have proposed several techniques to explain model predictions. However, practitioners struggle to use these explainability techniques because they often do not know which one to choose and how to interpret the results of the explanations. In this work, we address these challenges by introducing TalkToModel: an interactive dialogue system for explaining machine learning models through conversations. TalkToModel comprises 1) a dialogue engine that adapts to any tabular model and dataset, understands language, and generates responses, and 2) an execution component that constructs the explanations. In real-world evaluations with humans, 73% of healthcare workers (e.g., doctors and nurses) agreed they would use TalkToModel over baseline point-and-click systems for explainability in a disease prediction task, and 85% of ML professionals agreed TalkToModel was easier to use for computing explanations. Our findings demonstrate that TalkToModel is more effective for model explainability than existing systems, introducing a new category of explainability tools for practitioners. We release code a demo for the TalkToModel system at: https://github.com/dylan-slack/TalkToModel.

## 1 Introduction

Machine learning (ML) models are being deployed to make consequential decisions in several critical domains such as healthcare, finance, and law, due to their strong predictive performance. However, state-of-the-art ML models, such as deep neural networks, have also become more complex and, therefore, hard to understand. This dynamic poses challenges in real-world applications for the model stakeholders who need to understand why models make predictions and whether to trust the predictions. Consequently, there are an increasing number of techniques that explain the predictions of ML models. However, recent work suggests practitioners often have difficulty interpreting the results of the explanations and determining which ones to run [17, 14]. Being able to understand ML models through simple and intuitive interactions is a key bottleneck in adoption across many applications—particularly in settings where experts use models to aid decision making.

Natural language dialogues are a promising solution for supporting accessible interactions with ML models. However, designing a dialogue system that supports a satisfying model understanding

---

[*]Equal Advising

2022 Trustworthy and Socially Responsible Machine Learning (TSRML 2022) co-located with NeurIPS 2022.

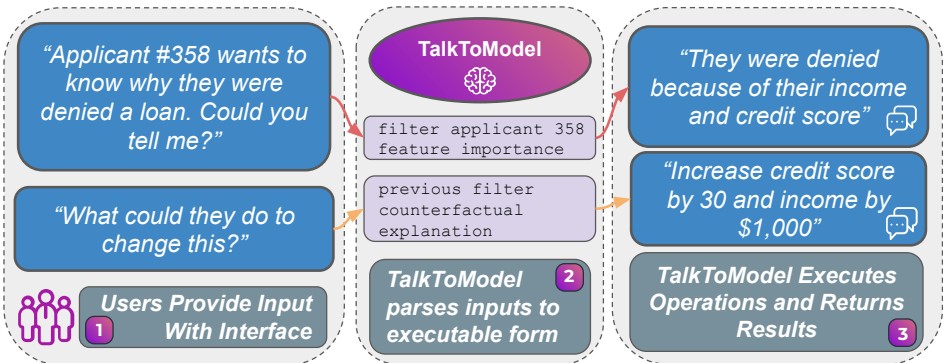

Figure 1: **Overview of TalkToModel:** **(1)** users supply natural language inputs. **(2)** the dialogue engine parses the input into an executable representation. **(3)** the execution engine runs the operations and the dialogue engine uses the results in its response.

experience introduces several challenges. First, the system must handle many conversation topics about the model and data [38]. Further, the system should handle conversations across different application areas [5]. For example, participants will use different terminology in conversations about loan prediction compared to disease diagnosis. Last, the dialogue system should generate accurate responses that address the core questions presented by the user [27, 42].

In this work, we address these challenges by introducing TalkToModel: a system that enables open-ended natural language dialogues for understanding ML models for any tabular dataset and classifier. Users can have discussions with TalkToModel about why predictions occur, how the predictions would change if the data changes, and how to flip predictions, among many other operations. For example, on a disease prediction task, users can ask "how important is BMI for the predictions?" or "so how would decreasing the glucose levels by ten change the likelihood of men older than twenty having the disease?" TalkToModel will respond by describing how, for instance, BMI is the most important feature for predictions, and decreasing glucose will decrease the chance by $20\%$. To support such rich conversations with TalkToModel, we introduce techniques for both language understanding and model explainability. We implement a *dialogue engine* that parses user text inputs (referred to as *user utterances*) into an SQL-like programming language designed for model understanding. To support the system adapting to any dataset and model, we introduce lightweight adaption techniques that train large language models (LLMs) to perform the parsing. Last, we introduce an *execution engine* that runs the parses. To reduce the burden of users deciding which explanations to run, we introduce methods to automatically select explanations for the user, by runniung many explanations and comparing their fidelities.

We conduct extensive quantitative and human subject experiments to determine the efficacy of TalkToModel. We assess TalkToModel's language understanding generalization capabilities across diabetes prediction, rearrest prediction, and loan default prediction tasks and find TalkToModel is quite effective at generalizing to new situations. Next, we compare TalkToModel with the most popular open source explainability dashboard in a series of user studies with two groups (1) healthcare professionals with minimal or no ML experience and (2) ML professionals with intermediate or more ML experience. In both cases, over $80\%$ of participants agreed TalkToModel was easier to use than the baseline dashboard, and 77.7% of healthcare workers and 69.2% of ML professionals agreed they were more confident using TalkToModel. Overall, these results indicate that TalkToModel is effective at facilitating model understanding using open ended natural language dialogues.

## 2  TalkToModel

In this section, we introduce TalkToModel. First, we describe the dialogue engine and discuss how it understands user inputs, maps them to operations, and generates text responses based on the results of running the operations. Second, we describe the execution engine, which runs the operations.[2]

---

[2]Code and a demo of the system are provided here: https://github.com/dylan-slack/TalkToModel

Table 1: Overview of the *operations* supported by TalkToModel, which are incorporated into the conversation to generate responses.

| operation, arguments, and description | operation, arguments, and description |
|---|---|
| **Data** | **ML** |
| `filter(dataset, feature, value, comp)`: filter by using value and comparison operator | `predict(dataset)`: Predictions on `dataset` |
| `change(dataset, feature, value, dir)`: Increases, decreases, or sets feature by `value` | `likelihood(dataset)`: Prediction probabilities on `dataset` |
| `show(list)`: Shows items in list in the conversation | `incorrect(dataset)`: Incorrect predictions |
| `statistic(dataset, metric, feature)`: Computes summary statistic for `feature` | `score(dataset, metric)`: Model score |
| `count(list)`: Length of list | **Conversational** |
| `and(op1, op2)`: Logical "and" of two operations | `prev_filter(conversation)`: Gets last filters |
| `or(op1, op2)`: Logical "or" of two operations | `prev_operation(conversation)`: Gets last non-filtering operations |
| **Explainability** | `followup(conversation)`: Respond to system followups |
| `explain(dataset, method)`: Feature importances | **Description** |
| `cfe(dataset, number)`: Gets `number` cfe's | `function()`: Overview of the system's capabilities |
| `topk(dataset, k)`: Top `k` most important features | `data(dataset)`: Summary of dataset |
| `important(dataset, feature)`: Importance of `feature` | `model()`: Description of `model` |
| `interaction(dataset)`: Interaction effects | `define(term)`: Defines `term` |
| `mistakes(dataset)`: Error patterns `dataset` | |

## 2.1 Parsing User Utterances

To understand the intent behind user utterances, the system learns to translate or *parse* them into logical forms. TalkToModel performs the following steps to accomplish this: **1)** the system constructs a grammar for the user-provided dataset and model, which defines the set of acceptable parses, **2)** TalkToModel generates (utterance, parse) pairs for the dataset and model, **3)** the system finetunes a large language model (LLM) to translate user utterances into parses, and **4)** the system responds conversationally to users by composing the results of the executed parse into a response.

**Grammar** To represent the intentions behind the user utterances in a structured form, TalkToModel relies on a grammar, which expresses user utterances in a structured form. Compared with approaches that treat determining user intentions in conversations as a classification problem [19, 4], our approach expresses compositions of operations and arguments that are combinatorially challenging in a prediction setting. This grammar consists of production rules that include the `operations` the system can run (an overview is provided in Table 1), the acceptable `arguments` for each operation, and the relations between operations.

**Supporting Context In Dialogues** User conversations naturally make references back to events earlier in the conversation (e.g., "what do you predict for *them*?"). However, current models only parse a single input, making hard to apply them in settings where the context is important. We introduce two operations: `previous filter` and `previous operation`, which look back in the conversation to find the last filter and last operation, respectively. In contrast with approaches that maintain the conversation state using neural representations [6], grammar operations allow for much more trustworthy and dependable behavior while still fostering rich interactions, which is critical for high-stakes settings. These operations also act recursively, enabling users to build complex parses in multiple steps, supporting natural conversations for users.

**Parsing Dataset Generation** We finetune an LLM to translate utterances into the grammar in a seq2seq fashion. We automate the finetuning of an LLM to parse user utterances into the grammar by generating a training dataset of (utterance, parse) pairs. Compared to dataset generation methods that use human annotators to generate and label datasets for training conversation models [10, 31], this approach is much less costly, while still being highly effective. We write an initial set of user utterances and parses, where parts of the utterances and parses are *wildcard* terms. TalkToModel enumerates the

wildcards with aspects of a user-provided dataset, such as the feature names or values, to generate a training dataset. For instance, the utterance "How many people have {NUMERIC_FEATURE} above {NUMERIC_VALUE}" would have the numeric features and numeric values in the data enumerated to generate further training utterances. We have already written the initial set of utterances and parses, so users only need to provide their dataset to setup the conversation.

**Semantic Parsing**    We compare two strategies for using pre-trained LLMs to parse user utterances into the grammar **1.)** few-shot GPT-J [37] and **2.)** finetuned T5 [29]. The few-shot approach does not require finetuning the LLM and is quicker to setup, which makes it easier for users to get started with the system. However, the finetuned T5 leads to improved performance and a better user experience overall. For the few-shot approach, we prompt the LLM using (utterance, parse) pairs, ordering the closest pairs according to cosine distance of embedding immediately before the user utterance because LLMs exhibit recency biases and consider the GPT-J 6B model [43, 37]. We also fine-tune pre-trained T5 models in a seq2seq fashion on our datasets. and consider T5 Small, Base, and Large.

**Generating Responses**    TalkToModel composes the results of the operations into a natural language response it returns to the user. The system generates these responses by filling in templates associated with each operation based on the results. Compared to approaches that generate responses using neural methods [32], this approach ensures the responses are trustworthy and do not contain useless information hallucinated by the system, which could be a very poor user experience.

## 2.2  Executing Parses

In this subsection, we provide an overview of the execution engine, which runs the operations necessary to respond to user utterances in the conversation.

**Feature Importance Explanations**    TalkToModel explains why the model makes predictions to users with feature importance explanations. We implement the feature importance explanations using *post-hoc* feature importance explanations. Post-hoc feature importance explanations do not rely on internal details of the model $f$ (e.g., internal weights or gradients) and only on the input data $\mathbf{x}$ and predictions $\mathbf{y}$ to compute explanations, so users are not limited to only certain types of models [30, 22, 16, 28, 18]. While there exists several post hoc explanation methods, each one adopts a different definition of what constitutes an explanation (e.g., LIME returns coefficients of a local linear model, while SHAP computes Shapley values) [15]. Consequently, we automatically select the *most faithful* explanation for users, unless a user specifically requests a certain technique. This metric is the mean absolute difference between the model's prediction on the original input and a fudged version on $\mathbf{m} \in \{0,1\}^d$ features,

$$\text{Fudge}(f, \mathbf{x}, \mathbf{m}) = \frac{1}{N} \sum_{n=1}^{N} |f(\mathbf{x}) - f(\mathbf{x} + \epsilon_n \odot \mathbf{m})| \tag{1}$$

where $\odot$ is the tensor product and $\epsilon \sim \mathcal{N}(0, I\sigma)$ is $N \times d$ dimensional Gaussian noise. To evaluate faithfulness for a particular explanation method, we compute area under the curve on the top-k most important features, thereby summarizing the results into a single metric. We compute faithfulness for multiple different explanations and select the one with the highest value. Practically, we consider KernelSHAP [21] and LIME [30] when choosing explanation methods, because they tend to be able to explain models predictions well.

**Additional Explanation Types**    Since users have explainability questions that cannot be answered solely with feature importance explanations, we also support counterfactual explanations and feature interaction effects. These methods support topics around *how* to get different outcomes and if features *interact* with each other during predictions, supporting a broad set of user queries. We implement counterfactual explanations using DiCE, which generates a diverse set of counterfactuals [23], and feature interaction effects using the partial dependence based approach from Greenwell et al. [12].

**Exploring Data and Predictions**    Because understanding models often requires inspection of model predictions, errors, and the data itself, TalkToModel supports a wide variety of data and model exploration tools. For example, TalkToModel provides options for filtering data and performing what-if analyses, supporting user queries that concern subsets of data or what would happen if data

points change. Users can also inspect model errors, predictions, prediction probabilities, compute summary statistics, and evaluation metrics for individuals and groups of instances.

Table 2: Exact Match Parsing Accuracy (%) for the 3 gold datasets, on the IID and Compositional splits, as well as Overall for T5 and few-shot GPT models. The fine-tuned T5 models perform significantly better than few-shot GPT-J, and T5 Large performed the best. These results demonstrate that TalkToModel can understand user intentions with a high degree of accuracy using the T5 models.

| | German | | | Compas | | | Diabetes | | |
|---|---|---|---|---|---|---|---|---|---|
| | IID | Comp. | Overall | IID | Comp. | Overall | IID | Comp. | Overall |
| KNN | 26.2 | 0.0 | 16.5 | 27.4 | 0.0 | 21.9 | 10.9 | 0.0 | 8.4 |
| GPT-J 6B | | | | | | | | | |
| 5-SHOT | 51.6 | 14.9 | 38.0 | 51.3 | 6.9 | 42.5 | 55.8 | 7.0 | 44.7 |
| 10-SHOT | 57.9 | 9.5 | 40.0 | 49.6 | 3.4 | 40.4 | 53.7 | 9.3 | 43.7 |
| T5 | | | | | | | | | |
| SMALL | 61.1 | 32.4 | 50.5 | 71.8 | 10.3 | 59.6 | 77.6 | 30.2 | 66.8 |
| BASE | 68.3 | **48.6** | 61.0 | 65.0 | 10.3 | 54.1 | **84.4** | 34.9 | 73.2 |
| LARGE | **74.6** | 44.6 | **63.5** | **76.9** | **24.1** | **66.4** | **84.4** | **51.2** | **76.8** |

# 3 Evaluation

We demonstrate the accuracy of the system's language understanding capabilities and show it effectively understands users in dialogues. Next, we perform a human study and verify users both prefer and are more effective using TalkToModel than traditional point-and-click systems.

**Language Understanding**

Here, we quantitatively assess the language understanding capabilities of TalkToModel by creating gold parse datasets and evaluating the system's accuracy at parsing this data.

**Gold Parse Collection** While we synthetically generate training data, we construct gold datasets to evaluate the parsing performance of our models across diabetes prediction, loan prediction, and recidivism prediction tasks to ensure we evaluate TalkToModel in diverse settings. We adopt an approach inspired by Yu et al. [41] to construct the datasets. First, we write $50$ (utterance, parse) pairs for the particular task (i.e., loan or diabetes prediction). From there, we ask Mechanical Turk workers to rewrite the utterances while preserving their semantic meaning. Revising in this way ensures that the ground truth parse for the revised utterance will stay the same, but the system will have a new evaluation utterance. We ask workers to rewrite each pair $8$ times for a total of $400$ (utterance, parse) pairs per task. Next, we filter out low-quality mturk revisions. We ask the crowd sourced workers to rate the similarity between the original utterance and revised utterance on a scale of (1-4), where $4$ indicates the utterances have the same meaning and 1 that they do not have the same meaning. We collect 5 ratings per revision and remove (utterance, parse) pairs that score below $3.0$ on average.

**Results** We present the results in Table 2. To evaluate performance on the datasets, we use the exact match parsing accuracy and provide results on the IID and compositional split of the data [36, 41, 13]. The IID split contains (utterance, parse) pairs where the parse's `operations` and their structure (but not necessarily the arguments) are in the training data. The compositional split consists of the remaining parses that are not in the training data. Because LM's struggle compositionally, this split is generally much harder for LM's to parse [24, 40]. The fine-tuned T5 performs better overall than the few shot GPT-J models. In particular, the T5 Large models perform strongly on both the IID and compositional data and can even parse complex compositional phrases. Notably, the T5 small model performs better than the GPT-J 6B model, which has two orders of magnitude more parameters. This dynamic is particularly true in the compositional splits in the data where the GPT-J few shot models never exceed $10\%$ parsing accuracy. Overall, these results indicate TalkToModel can understand user utterances with a high degree of accuracy using our best performing $T5$ models. Further, we recommend using this model for the best results and use it for our remaining evaluation.

Table 3: User study results: % of respondents that agree (> Neutral Likert score) TalkToModel is better than the dashboard in the 4 comparison questions. A significant portion of respondents agreed TalkToModel is better than the dashboard in all the categories except Grad. students and "Likeliness To Use". Still, a majority agreed TalkToModel was superior in this case.

|  | % Agree TalkToModel Better | |
| Comparison | Health Care Workers | ML Grad. Students |
| --- | --- | --- |
| Ease of Use | 82.2 | 84.6 |
| Confidence | 77.7 | 69.2 |
| Speed | 84.4 | 84.6 |
| Likeliness To Use | 73.3 | 53.8 |

**User Study: Utility of Explainability Dialogues**

In this subsection, we evaluate how well the end-to-end system enables users to understand ML models compared to current systems.

**Study Overview**    We compare TalkToModel against *explainerdashboard*, one of the most popular open-source explainability dashboards [8] and measure their experiences using both systems. This dashboard includes similar functionality to TalkToModel but provides access to explanations, predictions, and other operations through point-and-click interactions. We perform this comparison using the Diabetes dataset, and a gradient boosted tree trained on the data [26].

To compare both systems in a controlled manner, we ask participants to answer general ML questions with TalkToModel and the dashboard. Each question is about basic explainability and model analysis, and participants answer using multiple choice, where one of the options is "Could not determine." if they cannot figure out the answer (though it is straightforward to answer all the questions with both interfaces). For example, questions are about comparing feature importances "Is glucose more important than age for the model's predictions for data point 49?" or model predictions "How many people are predicted not to have diabetes but do not actually have it?" Participants answer 10 total questions. We divide the 10 questions into 2 blocks of 5 questions each. Both blocks have similar questions but different values to control for memorization (exact questions given in Appendix A). Participants use TalkToModel to answer one block of questions and the dashboard for the other block. In addition, we provide a tutorial on how to use both systems before showing users the questions for the system. Last, we randomize question, block, and interface order to control for biases due to showing interfaces or questions first.

**Metrics**    Following previous work on evaluating human and ML coordination and trust, we assess several metrics to evaluate user experiences [7, 9, 11]. We evaluate the following statements along 1-7 Likert scale at the end of the survey:

- **Ease of Use:** *I found the conversational interface easier to use than the dashboard interface*

- **Confidence:** *I was more confident in my answers using the conversational interface than the dashboard interface*

- **Speed:** *I felt that I was able to more rapidly arrive at an answer using the conversational interface than the dashboard interface*

- **Likeliness To Use:** *Based on my experience so far with both interfaces, I would be more likely to use the conversational interface than the dashboard interface in the future*

To control for bias associated with the ordering of the terms conversational interface and dashboard interface, we randomized their ordering in the questions. For example, users would either see *I found the conversational interface easier to use than the dashboard interface* or *I found the dashboard interface easier to use than the conversational interface*. In addition to these metrics, we measure accuracy and time-taken to answer each question. Last, we asked to participants to write a short description comparing their experience with both interfaces to capture participants qualitative feedback about both systems.

Table 4: User study results: Completion rate and accuracy across interfaces and participant groups. We compute the completion rate as the questions users provided and answer for and did not mark "could not determine." We measure accuracy on completed questions. Participants answered questions at a higher rate more accurately using TalkToModel than the dashboard.

|  | % Questions Completed | | % Accuracy On Completed Questions | |
| --- | --- | --- | --- | --- |
|  | Dash. | TalkToModel | Dash. | TalkToModel |
| Health Care Workers | 74.7 | 86.2 | 66.1 | 91.8 |
| ML Grad. Students | 73.8 | 93.9 | 62.5 | 100.0 |

**Recruitment**    Since TalkToModel provides a highly accessible way to understand ML models, we expect it to be useful for subject matter experts with variable degrees of ML expertise. As such, we recruited 45 English speaking healthcare workers to take the survey using the Prolific service [25]. This group comprises a range of healthcare workers, including doctors, pharmacists, dentists, psychiatrists, healthcare project managers, and medical scribes. As another point of comparison, we recruited ML professionals with relatively higher ML expertise from ML Slack channels and email lists. We received 13 potential participants, all of which had graduate course level ML experience or higher, and included all of them in the study. We received IRB approval for this study from our institution's IRB approval process.

**Metric Results**    A significant majority of health care workers agreed they preferred TalkToModel in all the categories we evaluated (Table 3). The same is true for the ML professionals, save for whether they were more likely to use TalkToModel in the future, where 53.8% of participants agreed they would instead use TalkToModel in the future. In addition, participants subjective notions around how quickly they could use TalkToModel aligned with their actual speed of use, and both groups arrived at answers using TalkToModel significantly quicker than the dashboard. The median question answer time (measured at the total time taken from seeing the question to submitting the answer) using TalkToModel was 76.3 seconds, while it was 158.8 seconds using the dashboard.

Participants were also much more accurate and completed questions at a higher rate (i.e., they did not mark "could not determine) using TalkToModel (Table 4). While both health care workers and ML practioners clicked could not determine for around a quarter of the questions using the dashboard, this was only true for 13.8% of health care workers and 6.1% of ML professionals using TalkToModel, demonstrating the usefulness of the conversational interface. Further, on the questions they completed, both groups were much more accurate using TalkToModel than the dashboard. Most surprisingly, though ML professionals agreed they preferred TalkToModel only about half the time, they answered all the questions correctly using it, while they only answered 62.5% of questions correctly using the dashboard. These results indicate TalkToModel is a much more effective system for understanding ML models for users with a variety of skill levels.

## 4   Conclusion

In this paper, we introduced TalkToModel, a modular platform for performing open-ended dialogues for understanding ML models. We showed how end-users prefer TalkToModel over existing systems, find it easier to use, and can understand their models more rapidly and accurately using the system. These results demonstrate that TalkToModel serves as a very effective route for providing explanations to end-users of all skill levels. In the future, it will be helpful to investigate applications of TalkToModel *in-the-wild*, such as in doctors' offices, laboratories, or professional settings, where model stakeholders use the system to understand their models.

## 5   Acknowledgments

The authors would like to thank the anonymous reviewers for their helpful feedback and all the funding agencies listed below for supporting this work. This work is supported in part by the NSF awards #IIS-2008461, #IIS-2008956, #IIS-2046873, #IIS-2040989, and research awards from Google,

JP Morgan, Amazon, Harvard Data Science Initiative, $D^3$ Institute at Harvard, and the HPI Research Center in Machine Learning and Data Science at UC Irvine. HL would like to thank Sujatha and Mohan Lakkaraju for their continued support and encouragement. The views expressed here are those of the authors and do not reflect the official policy or position of the funding agencies.

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

# A   Experimental Details

In this appendix, we provide an overview of additional experimental details. First, we provide additional details about the grammar. Next, we describe how we generate our parsing training sets. Further, we provide more details about the explanation selection procedure. After, we provide examples of the sentences we had mturk workers revise to create our gold parsing datsets. Finally, we provide additional details about the user study.

## A.1   Model Details

**Finetuning**   To perform fine-tuning for the T5 models, we split the dataset using a 90/10% train/validation split and train for 20 epochs to maximize the next token likelihood with a batch size of 32. We select the model with the lowest validation loss at the end of each epoch. We fine-tune with a learning rate of $1e$-4 and the AdamW optimizer [20]. and consider T5 Small, Base, and Large variants.

**Guided Decoding**   When parsing the utterances, one issue is that their generations are unconstrained and may generate parses outside the grammar, resulting in the system failing to run the parse and bad user experiences. To ensure the generations are grammatical, we constrain the decodings to be in the grammar [33]. This technique, referred to as *guided decoding*, constrains the LLM generations to only allow those tokens that appear next in the grammar at any point during generation. Practically, we accomplish this by recompiling the grammar at inference time into an equivalent grammar consisting of the tokens in the LLM's vocabulary. While decoding from the LLM, we fix the likelihood of ungrammatical tokens to $0$ at every generation step. Thus, the LLM only generates grammatical parses.

## A.2   Grammar Details

In this subsection, we provide additional details about the grammar. First, we describe the design of the grammar. After, we provide details about how we update the grammar for new datasets.

**Design**   Recall from the main paper that the grammar serves as a logical form of user utterances, which TalkToModel can execute. Here, we provide more details about the grammar design. The grammar defines relations between the `operations` and the acceptable values `arguments` in Table 1. For example, the grammar defines different acceptable values for the `comparison` argument in the `filter` operation, such as `less than or equal to` or `greater than`. In addition, we structure the grammar to make parses appear closer to natural language text instead of a formal programming language like SQL or Python. The reason is that language models tend to perform better at translating utterances into grammars that are more similar to natural language instead of a programming language [33]. Consequently, we design the grammar, so that parses appear more like natural language text, without unnecessary parentheses and commas, and omitting unnecessary arguments where possible. In general, we found that simplifying the grammar and making it more like natural language as much as possible considerably improved performance. For example, the question "What are the three most important features for people older than thirty-five?" would simply translate to `filter age greater than 35 and topk 3`. Note, here, because TalkToModel is applied to only one dataset at a time, the `dataset` argument can be omitted for simplicity. Practically, we implement the grammar in Lark [34] because the implementation supports interactive parsing, simplifying the process of implementing the guided decoding strategy.

**Updating the Grammar For Datasets and User Utterances**   In the main paper, we discussed how we update the grammar based on the dataset. Here, we provide more description about how we update the grammar. We update the grammar based on the feature names and categorical feature values in the dataset. In particular, the acceptable values for the `feature` argument (Table 1) in the grammar becomes all the feature values in the dataset. Further, the `value` argument for a categorical `feature` becomes the set of categorical feature values that appear in the data for that feature. Because there are many possible values of numeric features, we instead extract potential numeric values from user utterances as they are provided to the system. Specifically, we set the `value` argument in the grammar for numeric features to contain the set of numeric values that appear in the user utterance.

We additionally support string based numeric values (e.g., "fifty-five" or "twelve") to make the system handle a wider variety of cases.

### A.3 Training Dataset Generation

In this subsection, we provide details about the generation of the (utterance, parse) pair training dataset. To ensure that we generate a diverse and comprehensive set of (utterance, parse) pairs for training, we compose a total of $687$ templates that use $6$ different wildcard types. The templates consist of a diverse set of utterances that encompass the different operations permitted in the system. The wildcards include categorical feature names, numeric feature names, class names, numeric feature values, categorical feature values, explanation types, and common filtering expressions (e.g., "`{NUMERIC_FEATURE}` above `{NUMERIC_VALUE}`"). Because templates can have potentially many wildcards, we recursively enumerate all the wildcard values for each parse. Further, we also limit the number of values for certain wildcards to ensure the number of training pairs generated does not become extremely large. In particular, we limit the number of numeric values to $2$ values per feature. In addition, to prevent templates with many wildcards from dominating the training dataset, we also downsample the number of values per wildcard to $2$ values after the initial recursion. In this way, we the training dataset does not get dominated by a few templates that have many wildcards, which we found improves performance.

As an example, an utterance template is "Explain the predictions on data with `{NUMERIC_FEATURE}` greater than `{NUMERIC_VALUE}`" and the corresponding parse template is `filter {NUMERIC_FEATURE} greater than {NUMERIC_VALUE} and explain feature importance`. From there, we enumerate the numeric features in the dataset and a selection of numeric feature values, substituting these into `{NUMERIC_FEATURE}` and `{NUMERIC_VALUE}` respectively to generate data.

### A.4 Explanation Selection

Here, we provide additional details about the explanation selection algorithm. For our explanation selection algorithm, we set $\sigma = 0.05$, $K$ to floor$(\frac{d}{2})$, and $N = 10,000$. Further, because it would not make sense to perturb categorical features using Gaussian noise, we permute these features to another value occurring in the feature $30\%$ of the time. We additionally use both LIME [30] and KernelSHAP [21] with their default settings while computing explanations. In addition, we include LIME with the following kernel widths: $[0.25, 0.50, 0.75, 1.0]$, to generate explanations with a variety of different localities.

In addition, if the different in fidelity between the top two explanation methods is small, so it is hard to determine which explanation is most accurate, we use the explanation stability metric proposed by Alvarez-Melis and Jaakkola [3] to break ties, because it is more desirable for the explanation to robust to perturbations [35, 2]. In order to use the *stability* metric proposed by Alvarez-Melis and Jaakkola [3] to break ties if the explanations fidelities are quite close (less than $\delta = 0.01$), we compute the jaccard similarity between feature rankings instead of the $l2$ norm as is used in their work. The reason is that the norm might not be comparable between explanation types, because they have different ranges, while the jaccard similarity should not be affected. Further, we compute the area under the top k curve using the jaccard similarity stability metric, to make this measure more robust.

### A.5 Gold Parsing Dataset

In this subsection, we provide examples of 10 of the sentences provided to mturk workers to revise during our data generation process for each dataset. The examples are provided in Table 5 and illustrate the different types of utterances revised by mturkers to create a comprehensive testing set. Also, we provide the number of (utterance, parse) pairs in the IID and compositional splits in Table 6.

### A.6 User Study Questions

In this subsection, we provide the questions participants answered in the user study. The questions are provided in Table 7. One of the two question blocks, Block 1 or Block 2, is shown for TalkToModel and the other is shown for the dashboard (the ordering of TalkToModel and the dashboard is also

Table 5: Examples of 10 sentences out of 50 from each dataset provided to mturk workers to revise.

| | |
|---|---|
| COMPAS | What is your reasoning for determining if people older than 20 are likely to commit crimes? |
| | How likely are people that are younger than 25 or have committed at least 1 crime in the past to commit a crime in future? |
| | what are top 3 most important features you use for prediction for people if they were to decrease their prison terms by 10 months? |
| | For people that are 18 years old and black, how often are you correct in predicting whether they will commit crimes in the future? |
| | let's look at those in the data with 3 or more prior crimes on record. what are some common mistakes the model makes on these people? |
| | For this subset in the data, how accurate is the model? |
| | For people that are 18 years old and black, how often are you correct in predicting whether they will commit crimes in the future? |
| | how likely would the person with the id number of 33 in the data be to a commit a crime if they were 5 years younger? |
| | But what if they were twenty years older? |
| | Could you show me some data for people who are black? |
| Diabetes | How likely are people that either (1) have had two pregnancies or (2) are older than 20 and younger than 30 to have diabetes? |
| | What are the top five most important features for the model's predictions on people with a bmi over 40? |
| | Show the data for people older than 20. Then, could you show me the predictions on this data? |
| | what would happen to the likelihood of having diabetes if we were to increase glucose by 100 for the data point with id 33 |
| | What's the average age in the data? |
| | For this subset in the data, how accurate is the model? |
| | What does patient number 34 need to do in order to be diagnosed as unlikely to have diabetes? |
| | What are the reasons why the model predicted data point number 100 and what could you do to change it? |
| | How would the predictions change if age were decreased by 5 years for people with a bmi of 30? |
| | How do the features of the data interact in the model's predictions on this particular data? |
| German | If people in the data were unemployed, how important would the age and loan amount features be for predicting credit risk? |
| | what would happen to the likelihood of being bad credit risk if we were to increase the loan amount by 250 for the data point with id 89 |
| | what are top three most important features for determining whether those who are applying for furniture loans are good credit risk? |
| | What is the average loan amount for people with no current loans and that do not own a house? |
| | How accurate are you at predicting whether people who are asking for loans for home appliances are good credit risk? |
| | In the dataset, if the loan duration were to be increased by 2 years, what would the predictions for the data be? |
| | Could you show me some examples the model predicts incorrectly and how accurate the model is on the data? |
| | What do you predict on the instances in the data? Also, could you show me an example of a few mistakes you make in these predictions? |
| | But why did you think these people are bad credit risk? |
| | If these people were not unemployed, what's the likelihood they are good credit risk? Why? |

Table 6: The number of gold (utterance, parse) pairs in the IID and Compositional splits for the datasets. There are relatively more IID questions in each dataset.

| | COMPAS | Diabetes | German |
|---|---|---|---|
| IID | 117 | 147 | 127 |
| Compositional | 29 | 43 | 74 |
| Overall | 146 | 190 | 201 |

randomized). The two question blocks include similar concepts to ensure a similar level of difficulty but include different numbers to discourage memorization between the question blocks.

Table 7: The question's participants answered during the user study. Participants are shown either the questions in Block 1 or Block 2 for TalkToModel and the other set of questions for dashboard.

| | |
|---|---|
| Block 1 | What are the three most important features for the model's predictions on people older than 30 with bmi's above 35? |
| | What is the feature importance ranking of the age feature for data point id 188? |
| | How many individuals in the dataset are predicted to be likely to have diabetes but are not actually likely to have it? |
| | If patient id 293 were to decrease their bmi by five, what's the prediction probability of the "likely to have diabetes" class? |
| | Is the "glucose" feature more important than the "age" feature for the model's prediction on data point 49? |
| Block 2 | What are the three most important features for the model's predictions on people younger than 23 with glucose levels below seventy-five? |
| | What is the feature importance ranking of the insulin feature for data point id 57? |
| | How many individuals in the dataset are predicted not likely to have diabetes but actually are likely to have it? |
| | What's the likelihood of patient 57 having diabetes if they increased their glucose levels by 100 and bmi by 3? |
| | How important is the "diabetes pedigree function" feature compared to the "glucose" feature for the model's prediction on data point 55? |

## A.7  User Study Length & Compensation

The survey took around 30 minutes for participants to complete. We compensated the ML professionals by providing them with a $20 gift card. We paid the prolific workers $14.74/hour on average for completing the survey, considering their individual completion times.

## B  Additional Experiments

In this appendix, we provide additional experimental results. First, we provide experimental results where we show the advantages of explanation selection. Second, we give results on the affects of the

number of training templates. After, we provide error analysis for our parsing models. Last, we give additional user study results.

## B.1 Example Conversation

In this subsection, we provide an example dialogue using the system, in Table 8.

Table 8: A conversation about diabetes prediction, demonstrating the breadth of different conversation points the system can discuss.

| User "utterances" & TalkToModel response |
|---|
| "What are the predictions on patients older than 30? Could you also tell me what would happen to these predictions if we increased bmi by 10?" |
| On data with **age greater than 30.0**, the model predicts:
⇨ **No Diabetes** on 46.8% of data
⇨ **Diabetes** on 53.2% of data
On data with **age greater than 30.0 and bmi is increased by 10.0,** the model predicts:
⇨ **No Diabetes** on 30.1% of data
⇨ **Diabetes** on 69.9% of data |
| "So how important is bmi for the model's predictions on the over thirty group?" |
| For the model's predictions on data with **age greater than 30.0**, the **bmi** feature is ranked on average **2.86** (95% CI [2.685, 3.035]). Here, rank 1 is the most important feature (out of 8 features). Overall, **bmi** is **highly important** for the model's predictions. |
| "Ah, so which two features are mostly more important than bmi for these people?" |
| For the models predictions on data **where age is greater than 30.0**, the **top 2** most important features are:
1. **glucose**
2. **pedigree_function** |
| "What types of patients is the model typically predicting incorrect here"? |
| For data with **age greater than 30.0**, the model typically predict predicts incorrect:
⇨ if (**bmi > 26.95**) and (**glucose ≤ 125.0**) and (**pedigree_function**
    then the model is incorrect on *62.3%* of *53* samples
⇨ (**bmi > 26.95**) and (**glucose > 125.0**) and (**age > 57.5**)
    **<= 0.943**) then the model is incorrect on *62.5%* of *8* samples
⇨ (**bmi ≤ 26.95**) and (**glucose > 180.5**) then the model is incorrect
    on *100.0%* of *2* samples.
Want to take a closer look at these rules? |

## B.2 Advantages of Explanation Selection

In the main paper, we discussed how we select explanations based on those that have the highest fidelity in order to ensure the user receives the most accurate explanations possible. In this subsection, we provide more details about the advantages of explanation selection and show that without the procedure, users may receive comparatively lower fidelity explanations, demonstrating the importance of the technique.

Our evaluation procedure is as follows. For a data point, we compute the topk fidelity AUCs from all the explanations we consider (details about the explanation pool in Appendix A.4). Next, we compute several metrics that capture the improvement in fidelity using our selection technique. We compute the absolute improvement over the worst fidelity explanation, absolute improvement over the median

Table 9: The improvement in fidelity using our explanation selection procedure. We present the average value and (1 STD) for each metric across the 3 datasets. There are significant differences in the explanation fidelities, and the worst case explanation fidelity can be much worse than the one selected. These results demonstrate the importance of selecting explanations for end users, who may otherwise be left with comparatively lower fidelity explanations without this procedure.

|  | COMPAS | Diabetes | German |
|---|---|---|---|
| % Fidelity Decrease Using Median Fidelity Explanation | 54.9% (41.0) | 23.0% (20.7) | 5.0% (4.5) |
| % Fidelity Decrease Using Worst Fidelity Explanation | 68.5%(42.7) | 55.2% (31.8) | 57.1% (16.1) |
| Absolute Improvement Over Median Fidelity | 0.001 (0.002) | 0.035 (0.060) | 0.015 (0.012) |
| Absolute Improvement Over Worst Fidelity | 0.042 (0.071) | 0.002 (0.002) | 0.186 (0.08) |

fidelity explanation, percent decrease in fidelity using worst fidelity explanation, and percent decrease in fidelity using the median fidelity explanation. We perform this procedure for all the data points in a dataset.

We repeat this evaluation procedure for the three datasets we consider in the main paper: Compas, Diabetes, and German Credit. The results provided in Table 9 demonstrate that explanation fidelity can be much worse without selecting for explanations. For instance, if experts without sufficient data science experience were to run an explanation type arbitrarily, these results demonstrate they could have relatively much less faithful explanations.

### B.3 Effects of the number of training templates

In this subsection, we provide experimental details about the effects of the number of training templates on the parsing accuracy of the system. Because we use a template strategy to generate training data (Section 2), we must decide on how many prompts to write and include in the training scheme. This raises the question of how the number of templates affects model performance. To understand this behavior, we retrain the T5-Base model, randomly sampling the number of training templates over different percentages of the original templates set. In particular, we sweep over the following percentages $[20\%, 40\%, 60\%, 80\%, 100\%]$ on the diabetes dataset, downsampling and retraining 5 times per template. We give the results in Figure 2. We see that there are clear accuracy gains over using a smaller number of templates. Further, the gains for the compositional model performance seem to somewhat level off, but this is not the case for the IID split, suggesting further templates may help IID performance.

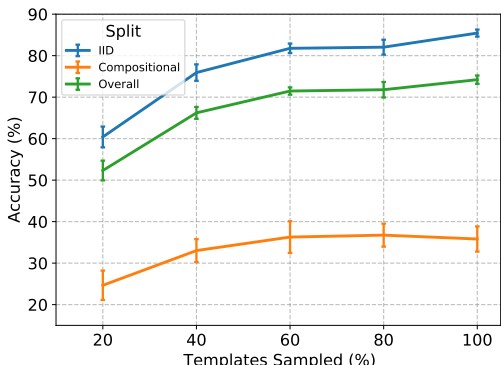

Figure 2: **Randomly sampling prompt templates and re-training T5-Base on the Diabetes dataset. For each down-sample %, the prompts are randomly down-sampled and the model is re-trained** 5 **times. The error bars are** 1 **standard deviation.**

### B.4 Parsing Error Analysis

In the main text, we demonstrated that the fine-tuned T5 models performed considerably better than few-shot GPT-J (Table 2). In this subsection, we perform additional error analysis for why this occurs.

|          | 5-Shot | | 10-Shot | |
|----------|------|---------------|------|---------------|
|          | IID  | Compositional | IID  | Compositional |
| Compas   | 29.8 | 77.8          | 15.3 | 53.6          |
| Diabetes | 40.0 | 40.0          | 19.1 | 28.2          |
| German   | 44.2 | 50.8          | 28.3 | 26.9          |

Table 10: Percentage of mistakes for few-shot GPT-J 6B where selected prompts *do not* include all the operations in the parse of the user utterance. We see that most of the time the operations for the parse of the user utterance are included in the prompts for the 10-Shot models, yet these methods still perform relatively poorly compared to finetuned T5.

This dynamic brings up the question: what is the cause of these poor few-shot results? Since the few-shot GPT-J models select the (utterance, parse) pairs from the synthetic dataset using nearest neighbors on a sentence embedding model, it could be possible these issues are due to the sentence embedding model failing to select good pairs. In particular, this nearest neighbors technique could fail to select pairs with the operations necessary to parse the user utterance, and not the model failing to learn from the examples. To evaluate whether this is the case, we compute the percent of mistakes that *do not* include the operations necessary to parse the user utterance for GPT-J. The results provided in Table 10 demonstrate that, especially for the 10-Shot case, the operations needed to parse the user utterance *are* included in the prompts, indicating the issues are likely due to the model's capacity to learn few-shot, rather than the selection mechanism. In this work, we were limited by using up to 6-billion parameter GPT-J, but it could be possible to achieve better results with larger models, as results on emergent abilitiees suggest [39].

## B.5 User Study Results: Per Question Likert Scores

In this subsection, we provide additional user study results. In addition to the questions asked at the end of the survey (Table 3 and Table 4), we also asked users to rate their experiences using both interfaces on a 1-7 Likert while they were taking the survey. In particular, we asked users how much they agreed with the following statements:

- I am confidence I completed my answer correctly.
- Completing this task took me a lot of effort.
- The interface was useful for completing the task.
- Based on my experience so far, I trust the interface to understand machine learning models.
- Based on my experience so far, I would use the interface again to understand machine learning models.

To evaluate these results, we compute the mean and standard deviation of the Likert score for the 1st through 5th question each user sees (question ordering is randomized so users see different questions first). We compute this for each statement and interface. The results for the medical workers are provided in Figure 3 and the ML professionals in Figure 4. Overall, the medical workers clearly prefer TalkToModel while answering the questions. Interestingly, they seem to gain trust in TalkToModel over time, going from "somewhat agree" to "agree" with the statement "Based on my experience so far, I trust the interface to understand ML models" by the end of the survey.

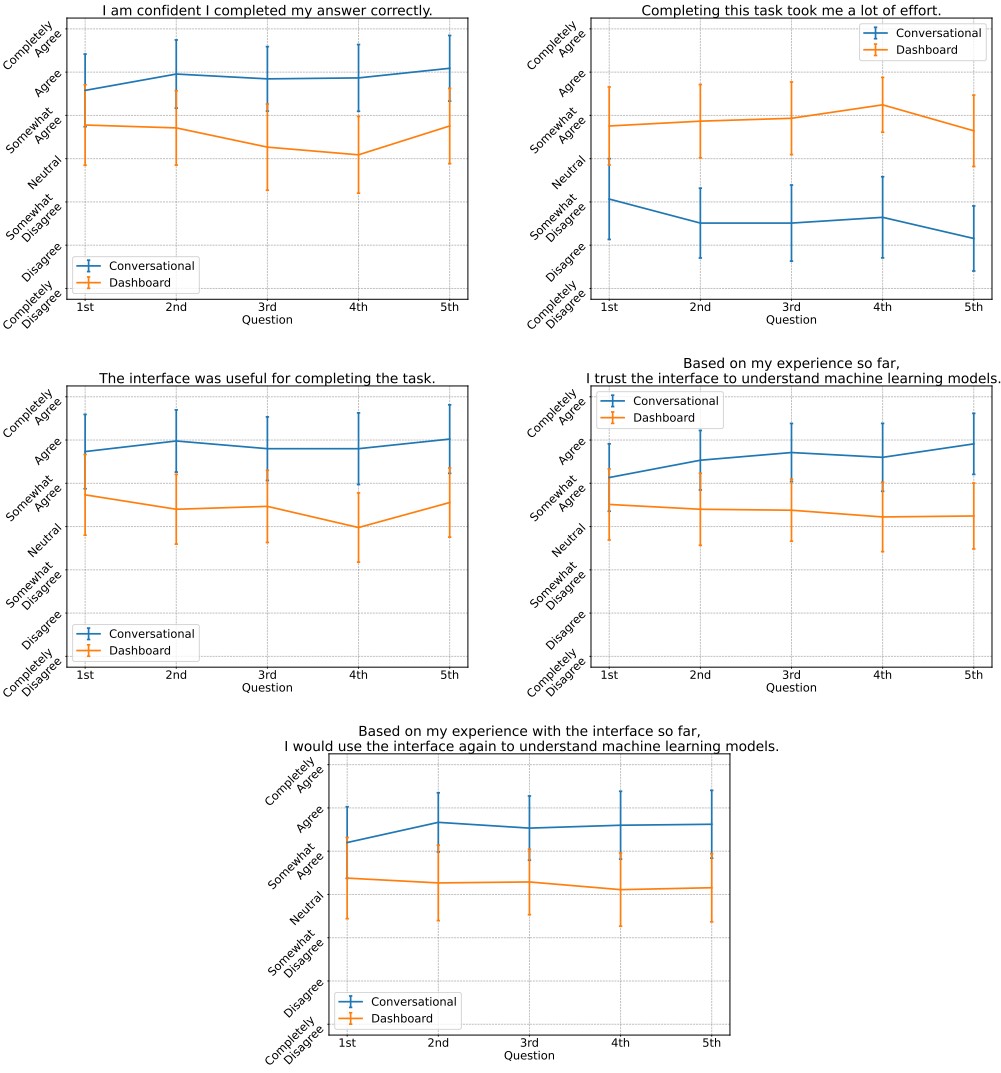

Figure 3: **Medical Worker per question likert results:** in general, these participants preferred TalkToModel over the dashboard to answer the questions.

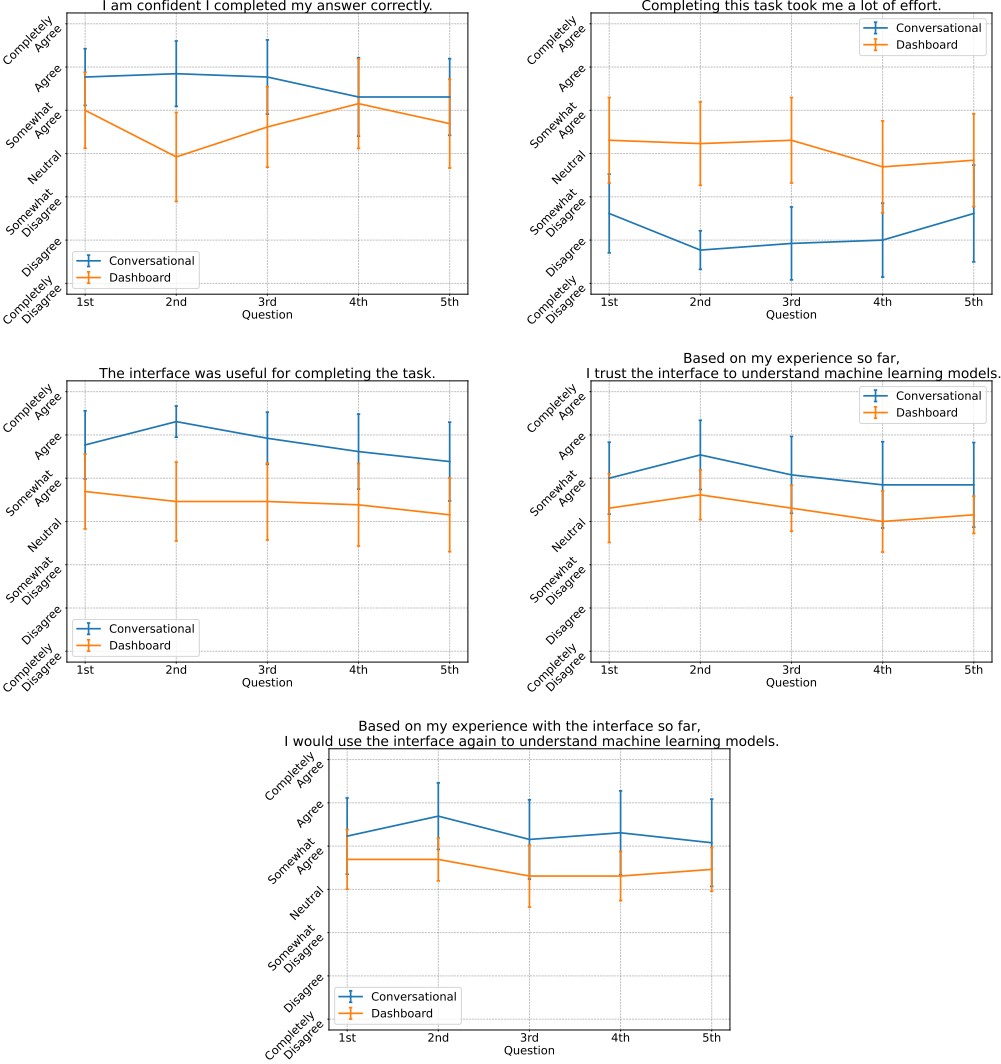

Figure 4: **ML professionals per question likert results:** these participants were more mixed about which interface they preferred while taking the survey. Interestingly, while the participants were much more accurate using TalkToModel, this population rated their answers at a similar confidence and said they trusted the interfaces similarly while taking the survey.

# C Interface Screenshots

In this appendix, we provide additional interface screen shots for both TalkToModel and the dashboard baseline.

## C.1 TalkToModel Interface

We provide additional screenshots of the TalkToModel GUI interface in Fig. 5 and Fig. 6. The interface is a text chat interface, where users provide input and the system response. On the right side of the interface, users can "pin" messages they find interesting. In addition, we provide a feature where users can get help to generate a question in a particular category (Fig. 7). This feature inputs a random synthetic training utterance into the command bar from the category of choice.

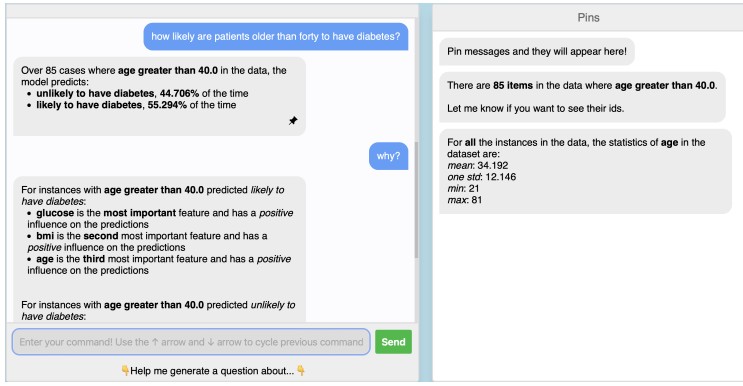

Figure 5: Discussing the model's predictions on patients older than $40$. The system correctly understand's the context of the question "why?".

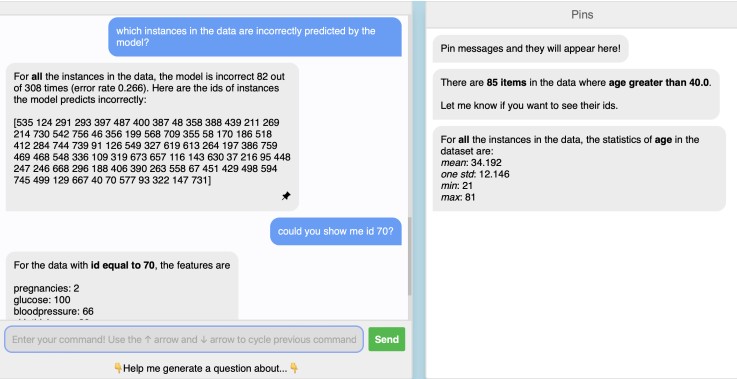

Figure 6: Beyond explanations, TalkToModel is also highly useful for classic error analysis, such as inspecting incorrectly predicted instances, as is shown here.

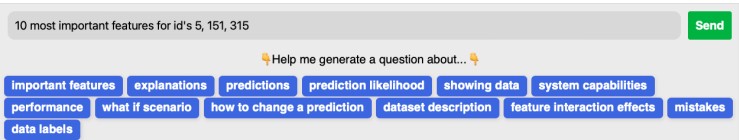

Figure 7: Screen shot of the help generate a question button, which will input a random question from the training data in the category when clicked..

## C.2 Dashboard Interface

We provide example screenshots for the baseline dashboard interface in Fig. 8 and Fig. 9 [8]. In depth description can be found in the project's documentation https://explainerdashboard.readthedocs.io/en/latest/. The dashboard provides different ways to understand ML models using point and click interactions.

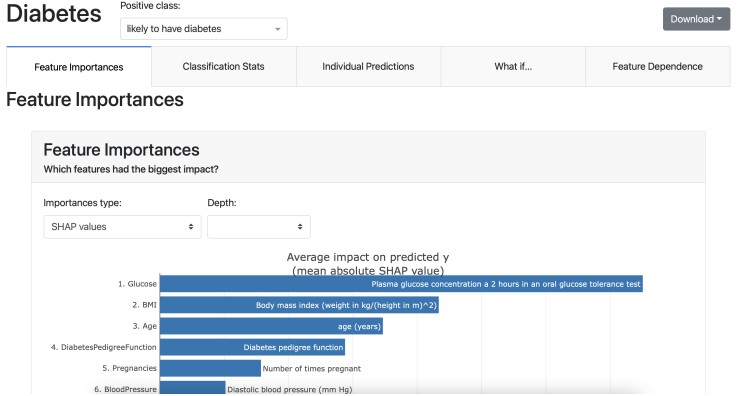

Figure 8: The dashboard interface lands on the global feature importances. Users can navigate to other tabs in the interface to see model predictions, metrics, feature importances, and compute what-if scenarios.

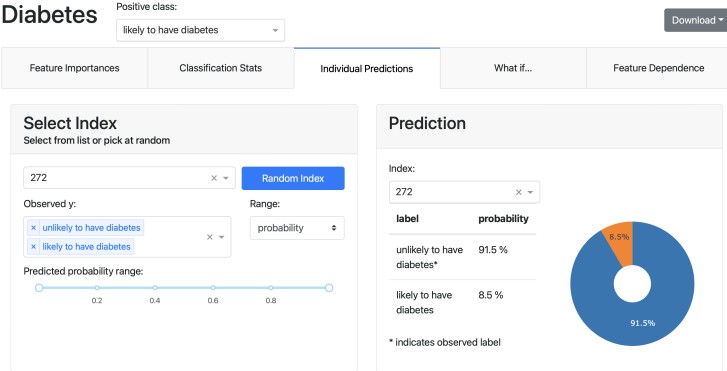

Figure 9: An example of the "individual predictions" page in the dashboard interface. Users need to navigate to this page and type in the data point index they want in order to inspect predictions.

