# OpenReview forum: "TalkToModel: Explaining Machine Learning Models with Interactive Natural Language Conversations"
_NeurIPS.cc/2022/Workshop/TSRML — TSRML2022_

### Official Review · Reviewer_bPYY · 2022-10-19
**An interesting work that takes an important next step towards explainable AI.**

**Overall Rating:** 9

**Summary:**

The paper introduces a dialogue system that can answer users' questions about the explanations of the models' predictions. It consists of a parse that translates users' input in natural language into executable operations, an execution engine that is responsible for different kinds of explaining algorithms, and finally a responser that returns the results to users in natural language.

The major contribution of the work is the parser which serves as a human-accessible interface between the users and the different explaining algorithms. To build such a parser, the work proposes to fine-tune an LLM with an automatic-generated training dataset. Another major contribution is defining a series of operations that support different kinds of explaining algorithms and recursive execution.

**Strengths:**

1. The proposed dialogue system for explaining the model's predictions to users is an important step towards explainable AI for a general audience. To achieve that goal, the work puts a substantial effort into making the system as user-friendly as possible (with natural language as both input and output).
2. The work designs a set of operations that can answer a wide range of questions about models' predictions.

**Weaknesses:**

Due to page limits, a large part of the technical details is moved to the appendix. Perhaps the authors can organize the main text so that it's more self-contained. For example, the part about constructing training data can be elaborated more.

**Overall Recommendation:**

I recommend accepting the paper given the importance of such a user-friendly system for explainable AI and the soundness of the proposed implementations.



**Review Confidence:**

4: The reviewer is confident but not absolutely certain that the evaluation is correct

---

### Official Review · Reviewer_ARpY · 2022-10-22
**Interesting problem and well written**

**Overall Recommendation:** It is well written, and the problem i…
**Overall Rating:** 6

**Summary:**

This paper proposes a TalkToModel framework that helps explain the machine learning models through conversations. The model consists of three parts: i) understanding text; ii) mapping them to operations; iii) and generates text responses. The model was evaluated both objectively by detecting user intensions and subjectively by user study. The experimental results on three databases as well as the subjective evaluation both validate the effectiveness.

**Strengths:**

The paper addresses the explainability in current machine learning models within a conversation, and it is relatively well written.

**Weaknesses:**

•	The paper exceeds six pages.

•	In addition to the explainability, how does this model compare to the most widely used transformer-based models in NLP? Also, what are the other benefits this model can offer compared to transformer-based models?


**Review Confidence:**

2: The reviewer is willing to defend the evaluation, but it is quite likely that the reviewer did not understand central parts of the paper

---

### Official Review · Reviewer_eF75 · 2022-10-22

**Overall Rating:** 8

**Summary:**

TalkToModel is a conversational framework to obtain prediction insights for a machine learning model. It is shown via user studies how end users can benefitted in solving a tasks (in a time bound fashion) with a help of a more natural avenue of interacting with a "so-called" black-box ML model.

**Strengths:**

1. The idea of conversational interpretations (even though post-hoc) is interesting and promising.
2. The paper uses light-weight off-the-shelf techniques for feedback parsing and implemented a grammar to restrict user actions for maximal benefit.
3. The final framework shows significant gain in usefulness in user studies.

**Weaknesses:**

It would be interesting to think how user can manipulate model predictions with feedback (think of feedback as critiques)

**Overall Recommendation:**

A strong paper, should be accepted. It motivates many interesting future directions.

**Review Confidence:**

5: The reviewer is absolutely certain that the evaluation is correct and very familiar with the relevant literature

---

### Decision · Program_Chairs · 2022-10-23

Accept